# Snoring Index and Neck Circumference as Predictors of Adult Obstructive Sleep Apnea

**DOI:** 10.3390/healthcare10122543

**Published:** 2022-12-15

**Authors:** Jui-Kun Chiang, Yen-Chang Lin, Chih-Ming Lu, Yee-Hsin Kao

**Affiliations:** 1Department of Family Medicine, Dalin Tzu Chi Hospital, Buddhist Tzu Chi Medical Foundation, Chiayi 622, Taiwan; 2Nature Dental Clinic, Nantou 545, Taiwan; 3Department of Urology, Dalin Tzu Chi Hospital, Buddhist Tzu Chi Medical Foundation, Chiayi 622, Taiwan; 4Department of Family Medicine, Tainan Municipal Hospital (Managed by Show Chwan Medical Care Corporation), Tainan 701, Taiwan

**Keywords:** apnea–hypopnea index (AHI), Epworth sleepiness scale (ESS), obstructive sleep apnea (OSA), snoring index (SI)

## Abstract

Background. Snoring is the cardinal symptom of obstructive sleep apnea (OSA). The acoustic features of snoring sounds include intra-snore (including snoring index [SI]) and inter-snore features. However, the correlation between snoring sounds and the severity of OSA according to the apnea–hypopnea index (AHI) is still unclear. We aimed to use the snoring index (SI) and the Epworth Sleepiness Scale (ESS) to predict OSA and its severity according to the AHI among middle-aged participants referred for polysomnography (PSG). Methods. In total, 50 participants (mean age, 47.5 ± 12.6 years; BMI: 29.2 ± 5.6 kg/m^2^) who reported snoring and were referred for a diagnosis of OSA and who underwent a whole night of PSG were recruited. Results. The mean AHI was 30.2 ± 27.2, and the mean SI was 87.9 ± 56.3 events/hour. Overall, 11 participants had daytime sleepiness (ESS > 10). The correlation between SI and AHI (*r* = 0.33, *p* = 0.021) was significant. Univariate linear regression analysis showed that male gender, body mass index, neck circumference, ESS, and SI were associated with AHI. SI (*β* = 0.18, *p* = 0.004) and neck circumference (*β* = 2.40, *p* < 0.001) remained significantly associated with AHI by the multivariate linear regression model. Conclusion. The total number of snores per hour of sleep and neck circumference were positively associated with OSA among adults referred for PSG.

## 1. Introduction

Obstructive sleep apnea (OSA) syndrome is a serious sleep disorder causing excessive daytime sleepiness (EDS). The severe consequences of OSA include coronary artery diseases, diabetes mellitus, ischemic stroke, hypertension, psychiatric disorders, and all-cause mortality [1,2]. The prevalence of OSA in adults has substantially increased over the last two decades; a community-based study reported an overall prevalence of approximately 26% for mild-to-severe sleep-disordered breathing (apnea–hypopnea index [AHI] ≥ 5) among individuals aged 30–70 years, and approximately 17% of men and 9% of women aged 50–70 years had moderate-to-severe sleep-disordered breathing (AHI ≥ 15) [3].

Polysomnography (PSG) is the gold standard for the diagnosis of OSA and monitoring of snoring [4]. Under the current guidelines, AHI continues to be referred to as the primary measure of severity of OSA [5]. Severity is evaluated as follows: normal, AHI < 5; mild, 5 ≤ AHI < 15; moderate, 15 ≤ AHI < 30; and severe, AHI ≥ 30 [6]. However, PSG requires technical expertise and is labor-intensive and time-consuming. Furthermore, timely access is a problem for many patients, and single-night PSG measurement might lead to misclassification of the severity of OSA. A previous study reported that 93% of women and 82% of men with moderate-to-severe OSA did not receive a diagnosis through PSG [7]. Another study reported that the severity of OSA varies every night due to the influence of body position and rapid eye movement on sleep, and that higher variability was observed in less severe OSA [8,9]. An ideal approach to overcome this uncertainty in the severity of OSA would involve repeated measures; however, this is often not feasible owing to the inconvenience caused to participants in undergoing repeated PSG tests in a hospital or laboratory [10].

Several methods, including questionnaires and objective measures, can help determine the probability that a patient has OSA. For instance, snoring is the cardinal symptom of OSA and has been reported in 70–95% of patients with OSA [4,11]. However, the correlation between snoring and OSA is not conclusive. One systemic review reported that snoring is a relatively accurate, but not a strong and reliable, method for diagnosing OSA [12]. In addition, a systematic review reported that the clinical symptoms of nocturnal gasping and choking are the most reliable indicators of OSA, whereas snoring is not very specific [13]. However, other studies reported strong associations between the severity of snoring and OSA [14]. Snoring index (SI, the total number of snores per hour of sleep) is one of the acoustic features of snoring sounds used to predict the AHI, and is positively correlated with the AHI [15,16,17,18].

EDS is a key symptom in many patients with OSA, and the Epworth sleepiness scale (ESS) is a simple and convenient self-report questionnaire used to assess EDS among patients with OSA in a clinical setting [19]. The ESS uses an eight-item scale, with each item scored from 0 through 3, and the total ESS score ranges from 0 to 24 [20]. ESS scores > 10 represent EDS, and patients with EDS need to be evaluated for potential OSA [21].

PSG involves recording various physiological signals during sleep; snoring is one of these signals. The purpose of this current study was to explore the association between SI and the severity of OSA, according to the AHI, among patients referred for PSG.

## 2. Materials and Methods

### 2.1. Ethical Considerations

The study protocol was reviewed and approved by the institutional review board of the Tainan Municipal Hospital (Managed by Show Chwan Medical Care Corporation) (SCMH_IRB No: 1090508) and the Research Ethics Committee of the Buddhist Dalin Tzu Chi Hospital, Taiwan (No. B10901020).

### 2.2. Materials

#### Data Collection from PSG

The participants underwent PSG recording using the Embla N7000 System (Embla Inc., Broomfield, CO, USA) shown in Figure 1. The recorded data provided various physiological signals, including electrocardiographic, electroencephalographic, electrooculographic, and electromyographic signals, and oxygen saturation and respiratory airflow. AHI is defined as the average hourly number of apnea and hypopnea events, which are categorized according to severity, where 5 ≤ AHI < 15 is mild, 15 ≤ AHI < 30 is moderate, and AHI ≥ 30 is severe [5].

An external piezo-electric snoring sensor (Embla N7000 Systems; attachable) can be used with the system. When a snoring sensor is attached to a patient’s throat, it generates a signal in response to the vibrations produced during snoring. Relying on vibrations rather than actual sounds eliminates all artifacts associated with external noises (the recording procedures conducted in this study had been obeyed strictly by the same operator to our established standard operation protocol). The snoring signal is converted to an analog voltage that can be measured. The SI was defined as the vibration times per sleep hour.

All respiratory signals from the Embla N7000 system were imported into the system in European Data Format, and a new anonymized polygraphy file was created for each patient. This software automatically validated sections where the signals of the pressure cannula, sound signal, and saturation were present [22]. The formal reports from the PSG included AHI and SI.

### 2.3. Method

#### 2.3.1. Design and Setting

Patients who reported snoring and were suspected of having OSA were referred for PSG test at a metropolitan regional teaching hospital in Taiwan between 1 January and 31 December 2021. Participants below 20 years of age were excluded. All patients provided written informed consent prior to enrolment.

#### 2.3.2. Study Outcome

Snoring index (SI, the total number of snores per hour of sleep) from PSG, ESS, and demographic factors were collected to evaluate the independent factors associated with the apnea–hypopnea index, as the measures of OSA severity.

### 2.4. Statistical Analysis

Statistical analyses were conducted using R software version 4.0.3 (R Foundation for Statistical Computing, Vienna, Austria). Two-sided *p* values < 0.05 were considered statistically significant. Continuous data are expressed as mean ± standard deviation (SD), and categorical variables are represented by frequency and percentage. The multivariate linear regression model was used to evaluate the independent factors associated with AHI.

## 3. Results

In total, 6 (12%) participants had primary snoring (AHI < 5), and 44 participants had OSA (AHI ≥ 5). The mean age of the participants was 47.5 ± 12.6 years, mean BMI was 29.2 ± 5.6 kg/m^2^, and mean neck circumference was 40.6 ± 5.3 cm. The mean SI was 87.9 ± 56.3 events/h, and the mean AHI was 30.2 ± 27.2. Eleven (22%) participants had EDS (ESS > 10). Additionally, values of BMI, neck circumference, and ESS were all higher in the AHI ≥ 5 group than the AHI < 5 group, which also reached statistical significance (*p* = 0.018, *p* = 0.013, and *p* = 0.037, respectively) (Table 1). The audio data handling and analysis was shown in Figure 2.

The mean SI for primary snoring (AHI < 5) was 89.28 ± 30.72 events/hour, and for mild, moderate, and severe OSA, it was 71.64 ± 48.07, 75.90 ± 39.99, and 106.30 ± 71.76 events/hour, respectively. No significant difference was observed in the SI by AHI score between the stratified four groups (*p* = 0.187) using ANOVA analysis (Table 2).

One reason for this might be the smaller number in each stratified group (*n* = 6, 14, 11 and 19, respectively), whereas the other might be that a higher mean value of SI was observed in the first (or fourth) group than either the second or third group. However, if we focused on the correlation between total SI and total AHI, a positively significant correlation (*r* = 0.33, *p* = 0.021) was demonstrated, and the correlation between the ESS and AHI was also significant (*r* = 0.35, *p* = 0.012). However, the correlation between the ESS and SI was not significant (*r* = 0.05, *p* = 0.717). The correlation between snoring index and apnea–hypopnea index is shown in Figure 3, and the correlation between Epworth sleepiness scale and apnea–hypopnea index is shown in Figure 4. Univariate analysis showed that gender, BMI, neck circumference, ESS, and SI were significant predictors of AHI (Table 3). Notably, a significantly positive correlation was found between neck circumference and BMI (*r* = 0.71, *p* < 0.001). Accordingly, collinearity of neck circumference and BMI has been concurred (both neck circumference and BMI were significant variables for AHI). Therefore, we shall only take one of the above variables (neck circumference or BMI) into account per multiple linear regression analysis (Model 1 or Model 2). Model 1 was better than Model 2 according to the adjusted R^2^ value (0.291 vs. 0.215). For each multiple linear regression analysis, only SI and neck circumference were significantly associated with AHI (Table 4).

## 4. Discussion

In the current study, SI and neck circumference were positively significantly correlated with AHI among adults referred for PSG. We suggested that the neck circumference for OSA might be a better parameter than BMI and ESS for OSA prediction.

The correlation between snoring and AHI has often been discussed in previous studies [10,12,13,14,15,16,17,18]. However, the findings have been inconclusive. One of the reasons for this might be the difficulty in the detection and identification of snoring. For example, Levartovsky et al. found the correlation between SI and AHI to be nonsignificant [18]. In that study, the acoustic signals were recorded using a noncontact directional microphone (RØDE, NTG-1, Silverwater, NSW, Australia) placed 1 m above the bed and connected to a digital audio recording device (Edirol R-4 Pro, Bellingham, WA, USA). Snoring was defined as breathing sound intensity > 50 dB, and SI was the total number of snores per hour of sleep. Wu found the correlation between SI and AHI to be significant [17]. In that study, the snoring burst index was defined as the number of snoring burst signal groups per hour of sleep. A burst signal group was defined as ≥3 successive bursts of signals with amplitudes exceeding the mean amplitude of all overnight snoring signals. In the current study, snoring measured through PSG and SI was defined as snoring events/h. The snoring sensor was attached to the patient’s throat, and a signal was generated in response to the vibrations that occurred during snoring. The snoring signal was then converted to an analog voltage that could be measured. We calculated the SI (total number of snores per sleep hour) from PSG data and found that the SI was correlated with AHI. The association between snoring and OSA according to the AHI is a very interesting issue for the study population and the investigators. Further study is warranted to investigate the detection of SI at home using similar measurements from PSG.

The ESS is often used clinically to assess subjective daytime sleepiness or sleep propensity in adults. The association between ESS and OSA has frequently been discussed, but no conclusion has been reached. Previous studies reported that ESS had only a poor to fair discriminatory ability to screen for OSA [20,23,24,25,26,27]. One study reported that ESS was associated with AHI among both men and women [28]. Moreover, another study reported that higher ESS item scores indicated a closer relationship with the corresponding AHI [29]. One of the possible explanations is the participant’s situational sleepiness, which influenced the results of the ESS [30]. In the current study, we found that although the correlation between ESS and AHI was significant for both men and women, it became nonsignificant after adjustments. This might be because ESS was associated with neck circumference and became nonsignificant, whereas neck circumference was a positive predictor of AHI. A previous cohort study reported that neck circumference was associated with ESS score among men [31].

Neck circumference, a proxy for upper body fat, has been shown to be associated with the risk factor for cardiovascular disease and metabolic syndrome [32,33]. In the current study, neck circumference was associated with AHI, and the mean neck circumference of middle-aged participants was 40.6 ± 5.3 cm. A previous study reported that neck circumference was a significant predictor of OSA; the neck circumference of the participants in that study was ≥17 inches [34].

A previous systemic review reported that BMI ≥ 25 kg/m^2^ was an important risk factor for OSA [35]. Another study showed that OSA was more common and more severe in Far East Asian men than in white men, although the value of mean BMI was lower in Far East Asian men (26.7 ± 3.8 kg/m^2^) than in white men (29.7 ± 5.8 kg/m^2^) [36]. In the current study, we found that BMI (29.2 ± 5.6 kg/m^2^) was a positive factor for AHI (*p* = 0.001) in univariate linear regression. A similar result was obtained if we used the factor of BMI to replace the neck circumference in multivariate linear regression analysis. The present study has some limitations. First, the sample size was small; only 50 participants (38 males; 12 females) were enrolled for analysis. Second, the sleep position of the participants could have affected the snoring signals. Third, the quality of the recorded signals was affected by sleep talk and sleep quality during the night. Finally, we found that individuals with primary snoring (AHI < 5) had higher SI than those with mild-to-moderate OSA (5 ≤ AHI < 30). One of the reasons could be the small number of participants (n = 6) diagnosed as having primary snoring. Further studies with more participants are warranted for a detailed investigation and to confirm our findings.

## 5. Conclusions

The mean number of snores per hour obtained from PSG and neck circumference were positively significantly correlated with AHI among adults. Further prospective and large-sample studies are warranted to confirm our results.

## 6. Patents

Predictors for OSA included snoring, male gender, older age, increased neck circumference, BMI, ESS, and a history of witnessed apneas [37]. In the current study, we suggested that the neck circumference for OSA might be a better parameter than BMI and ESS for OSA prediction.

## Figures and Tables

**Figure 1 healthcare-10-02543-f001:**
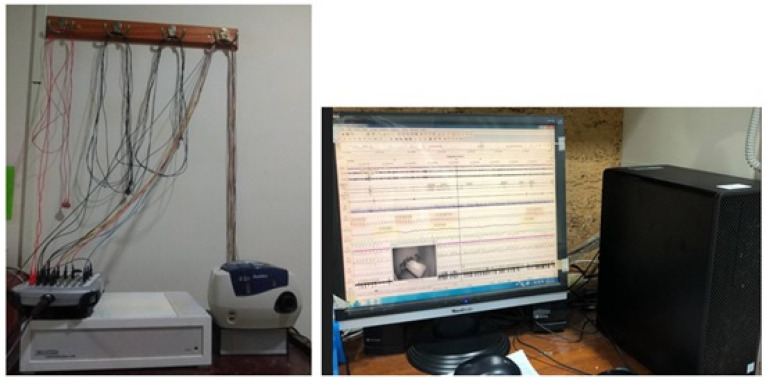
PSG recording device: the Embla N7000 System.

**Figure 2 healthcare-10-02543-f002:**
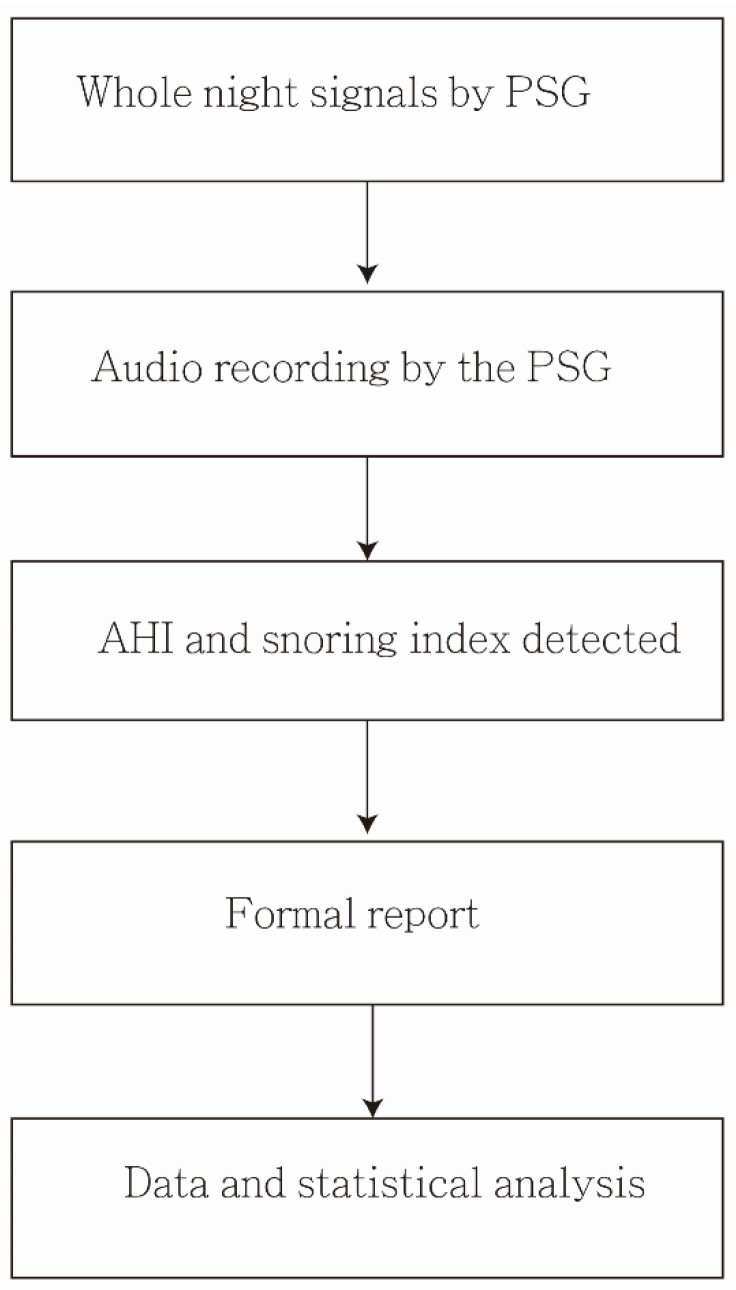
Block diagram of audio data handling and analysis.

**Figure 3 healthcare-10-02543-f003:**
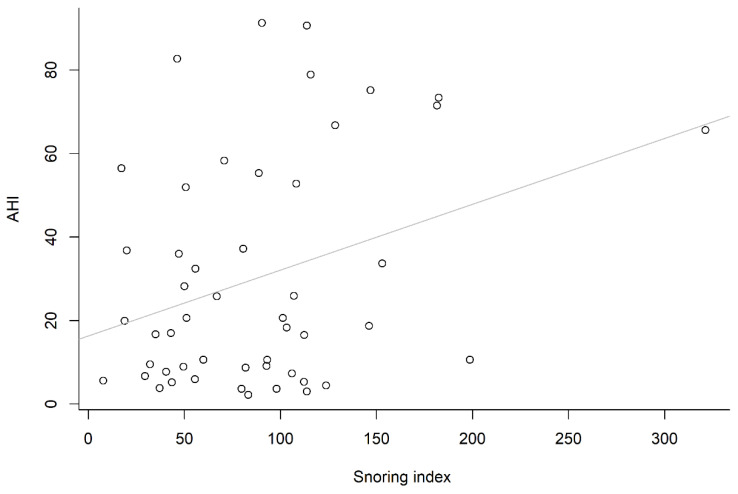
Positive correlation between snoring index and apnea–hypopnea index.

**Figure 4 healthcare-10-02543-f004:**
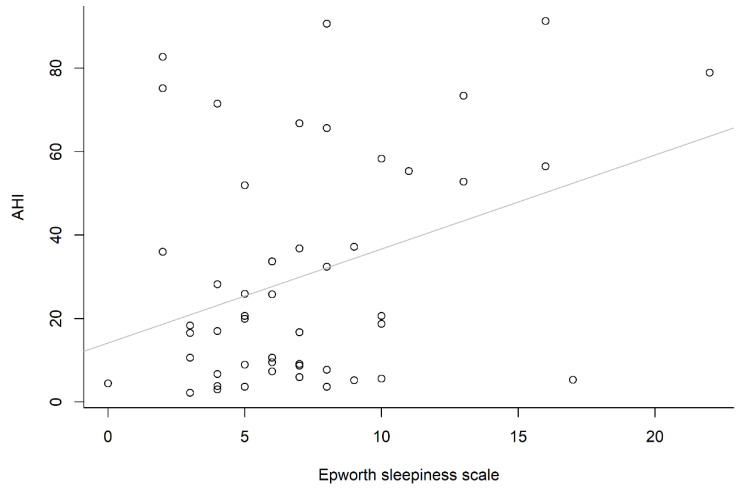
Positive correlation between Epworth sleepiness scale and apnea–hypopnea index.

**Table 1 healthcare-10-02543-t001:** Demographic characteristics of participants.

Variables	Value	AHI < 5	AHI ≥ 5	*p*
n (male/female)	50 (38/12)	16 (4/12)	44 (34/10)	0.621
Age, year	47.5 ± 12.6	47.8 ± 9.6	45.4 ± 13.0	0.663
BMI	29.2 ± 5.6	24.5 ± 4.3	29.8 ± 5.5	0.018
Neck circumference, cm	40.6 ± 5.3	36.5 ± 2.4	41.2 ± 5.3	0.013
Hypertension, yes	8	0	8	1
Diabetes, yes	0	0	0	-
Operation for snore	6	1	5	0.363
Education level *				0.217
Graduate school	7	0	7	
College	20	4	16	
others	12	0	12	
ESS (score > 10) [daytime sleepiness]	7.1 ± 4.3	4 ± 2.6	7.5 ± 4.3	0.037
Snoring index, events/sleep hour	87.9 ± 56.3	89.3 ± 30.7	87.7 ± 59.1	0.633
AHI score	30.2 ± 27.2	3.4 ± 0.8	33.8 ± 27.1	<0.001

Abbreviation: AHI, apnea–hypopnea index; BMI, body mass index; ESS, Epworth sleepiness scale. Education level *, missing data n = 11.

**Table 2 healthcare-10-02543-t002:** Distribution of snoring index by the apnea–hypopnea index score.

	AHI < 5	5 ≤ AHI < 15	15 ≤ AHI < 30	AHI ≥ 30	*p*
N	6	14	11	19	
Snoring index *	89.28 ± 30.72	71.64 ± 48.07	75.90 ± 39.99	106.30 ± 71.76	0.187

Snoring index *: snoring events per sleep hour.

**Table 3 healthcare-10-02543-t003:** Univariate linear regression for the snoring index, Epworth sleepiness scale, and apnea–hypopnea index.

	Snoring Index		ESS		AHI	
Variable	*β*	*p*	*β*	*p*	*β*	*p*
Male vs. female	4.39	0.816	1.15	0.424	18.86	0.035
Age	−0.23	0.723	−0.06	0.245	−0.31	0.317
BMI (kg/m^2^)	2.26	0.114	0.22	0.045	2.13	0.001
Neck circumference (cm)	−1.05	0.499	0.28	0.015	2.21	0.002
ESS	0.691	0.717			2.25	0.012
Snoring index	-		0.004	0.717	0.16	0.021
AHI	0.67	0.021	0.06	0.012		

Abbreviation: AHI, apnea–hypopnea index; BMI, body mass index; ESS, Epworth sleepiness scale.

**Table 4 healthcare-10-02543-t004:** Multiple linear regression for the apnea–hypopnea index.

Covariates	Model 1	Model 2	Model 3	Model 4	Model 5	Model 6
Intercept	−83.22	−34.37	−81.52	−78.56	−73.09	−69.40
	(0.003)	(0.066)	(0.005)	(0.021)	(0.032)	(0.039)
Snoring index, events/sleep hour	0.18	0.12	0.17	0.17	0.15	0.14
	(0.004)	(0.072)	(0.012)	(0.013)	(0.033)	(0.043)
Neck circumference, cm	2.40		2.16	2.20	1.46	1.05
	(<0.001)		(0.029)	(0.033)	(0.222)	(0.381)
Body mass index		1.86	0.31	0.23	0.77	0.83
		(0.005)	(0.735)	(0.830)	(0.505)	(0.465)
Age				−0.05	0.03	0.05
				(0.868)	(0.993)	(0.860)
Gender male vs. female					10.88	11.29
					(0.239)	(0.215)
ESS						1.30
						(0.123)
Adjusted R-squared	0.291	0.215	0.277	0.261	0.268	0.292

## Data Availability

The datasets generated during and/or analyzed during the current study are not publicly available, but are available from the corresponding author on reasonable request.

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
