# Peer review of "Snoring Index and Neck Circumference as Predictors of Adult Obstructive Sleep Apnea"

_healthcare, 2022, doi:10.3390/healthcare10122543_

Round 1

Reviewer 1 Report

Originality / Novelty of the study is low and sample size is to small for sufficient analysis and clear conclusions.

Manuscript is well written, but as per my suggestion, the main study limitation is sample size, so I am suggesting authors to include more patients in the study and to do new statistical analysis with new data, than results will be more reliable. I am suggesting to increase sample size for double.
Kind regards

Author Response

Please see the attachment. Many thanks for your kind suggestion and review.

Reviewer 2 Report

This is a scientific study, however, it is already well known that snoring and neck circumference are related to OSA. The authors should present the difference of their study from the previous reports, to be published in the journal.

Line 66 - Excessive daytime sleepiness (EDS) is repeated.

Line 119 – There is no subsection.

Line 131 – No significant difference was observed between which groups ? Isn’t the correlation between SI and AHI significant ?

In Table 4, other variables are missing. Although not significant, please present the values in the table.

Line 155 - Neck circumference is a better parameter than BMI is not scientific. It is well known that BMI is strongly related to OSA. If you want to say this, you should compare the two parameters statistically.

The manuscript needs English proofreading. Please don’t use “ Another study ~” repeatedly.

Author Response

(The authors gave the same response as above.)

Reviewer 3 Report

Ref.: healthcare-2024229

Article Title: Snoring Index and Neck Circumference as Predictors of Adult

 The purpose of this manuscript was to explore the association between SI and the severity of OSA, according to the AHI, among patients referred for PSG.

Reviewer’s comment:

1.     Please provide the picture of digital audio recording device.

2.     Does the same operator attach the snoring sensor to patient’s throat? Or is there any inter-observer operating error?

3.     In the paragraph of [Results], please provide the data (eg. mean and SD…) of BMI, Neck circumference, ESS, Snoring index by the two groups (AHI< 5, AHI ³ 5)

4.     In the paragraph of [Discussion], the present study has some limitations. First, the sample size was small… Is the sample size calculation needed in this kind of study??

Author Response

(The authors gave the same response as above.)

Round 2

Reviewer 1 Report

Dear authors,

Thank you for your answer.

Manuscript should be accepted for publication without any additional changes.

Kind regards

Author Response

Please see the attachment, thank you for your kind review!

Reviewer 2 Report

The authors did a lot of work to answer the questions successfully.  

Line 66 - Excessive daytime sleepiness (EDS) is repeated.

Response:

Thank you for your advice!

We have revised the manuscript as below:

We have deleted the redundant “(EDS)” in line 66 as suggested.

-> You can use the abbreviated word once you described in the manuscript.

Line 131 – No significant difference was observed between which groups? Isn’t the correlation between SI and AHI significant ?

Response:

Thank you for your advice!

If we stratified the subjects into four groups by levels of AHI, there showed no significant differences between the stratified four groups by ANOVA analysis (Table 2). The reason was that higher mean values were observed in the first (or fourth) group than either the second or the third group. However, if we focused on the correlation between total AHI and total snoring index, there demonstrated a positively significant correlation (Table 3 and Figure 2).

-> You can add this explanation to the manuscript because someone can be confused, and it seems that there is no correlation between SI and AHI.

Author Response

(The authors gave the same response as above.)
